# The Recurrence of Systemic Diseases in Kidney Transplantation

**DOI:** 10.3390/jcm14082592

**Published:** 2025-04-09

**Authors:** Gabriella Moroni, Marta Calatroni, Claudio Ponticelli

**Affiliations:** 1Department of Biomedical Sciences, Humanitas University, Via Rita Levi Montalcini 4, 20072 Milan, Italy; marta.calatroni@hunimed.eu; 2Nephrology and Dialysis Division, IRCCS Humanitas Research Hospital, Via Manzoni 56, 20089 Milan, Italy; 3Independent Researcher, 20131 Milan, Italy; ponticelli.claudio@gmail.com

**Keywords:** kidney transplantation, kidney survival, lupus nephritis, ANCA-associated vasculitis, ANCA-associated glomerulonephritis, Henoch–Schönlein purpura, IgA vasculitis, recurrent diseases

## Abstract

Kidney transplantation is the most effective replacement therapy for kidney failure, providing the best outcomes in terms of patient survival and offering a better quality of life. However, despite the progressive improvement in kidney survival, the recurrence of original disease remains one of the most important causes of graft loss and a major challenge that requires clinical vigilance throughout the transplant’s duration. Additionally, the type and severity of recurrence affect both treatment options and graft survival. This is especially true for the recurrence of systemic diseases. In this narrative review, we will discuss the timing, frequency, severity, and treatment of post-transplant recurrence in three systemic diseases: lupus nephritis (LN), Antineutrophil Cytoplasmic Antibodies (ANCA)-associated glomerulonephritis (ANCA-GN), and Henoch–Schönlein purpura (HSP). The recurrence of lupus nephritis is less common than that of primary focal segmental glomerulosclerosis or C3 glomerulopathy. Its severity can range from mild mesangial to diffuse proliferative forms, with varying prognoses and treatment options, much like the original disease. In some patients with LN, as well as in those with ANCA-GN or HSP, the reactivation of the primary disease can affect other organs besides the kidneys, potentially leading to life-threatening conditions. These cases may require a multidisciplinary approach, making these transplants clinically more challenging. Extrarenal flare-ups often necessitate an increase in immunosuppression, which in turn raises the risk of infections. In these autoimmune diseases, the role of immunological tests in determining the timing of kidney transplants remains a topic of ongoing debate. However, elevated levels of certain immunological markers, such as anti-dsDNA antibodies, ANCA titers, or serum immunoglobulin A may indicate a reactivation of the disease, suggesting the need for more intensive patient monitoring.

## 1. Introduction

Although it is reported that transplant recipients with recurrent disease are twice as likely to lose their allografts in comparison to those with no recurrence [1], the prevalence of recurrence is not precisely known for different reasons. The likelihood of recurrence depends on different factors such as the type of the original disease, the number of patients included in the cohorts, and the type of cohorts (from single-center or national or international transplant registries). The duration of the observation is particularly important, as some recurrence can develop even after 10 years of follow-up [2,3]. The different indications for performing graft biopsy (protocol biopsies vs. serial biopsies) and the number/types of procedures used for histological analysis also affect the results. The diagnosis of recurrence may require the examination of the graft biopsy using light microscopy and immunofluorescence, as well as electron microscopy, which is not routinely performed. This is especially important when ruling out transplant complications such as transplant glomerulopathy. With these caveats in mind, in this narrative review, we will discuss the recurrence of grafts in three systemic diseases (Systemic Lupus Erythematous, Antineutrophil Cytoplasmic Antibodies (ANCA)-associated vasculitis, Henoch–Schönlein purpura), primarily based on the recent literature.

### Materials and Methods

We carried out a comprehensive literature search of the last 20 years using the following terms: kidney transplantation; kidney survival; lupus nephritis; ANCA-associated vasculitis, ANCA-associated glomerulonephritis; Henoch–Schönlein purpura; IgA vasculitis; and recurrent graft diseases. Our search was performed in databases including PubMed, Medline, and Embase, as well as through the reference lists of the retrieved articles. Additionally, we manually searched the cited papers to identify additional studies relevant to the topic. The quality of the studies was assessed based on criteria including the number of participants and the importance of the published findings.

The purpose of this study is to review the timing, frequency, severity, and treatment of post-transplant recurrence in lupus nephritis (LN), ANCA-associated glomerulonephritis (ANCA-GN), and Henoch–Schönlein purpura (HSP), highlighting their impact on patient and graft survival and the challenges they pose in clinical management.

## 2. Kidney Transplant in Lupus Nephritis

LN occurs in thirty to fifty percent of patients within the first ten years of diagnosis of Systemic Lupus Erythematosus (SLE). It is one of the most severe manifestations of SLE and is associated with an increased risk of morbidity and mortality [4,5]. Kidney biopsy allows patients with LN to be classified into six classes, which are associated with different severities of presentation and prognosis [6]. The most severe histological forms of LN (class III and class IV LN) are the most frequently observed and require aggressive therapy to avoid the development of chronic kidney disease and end-stage kidney disease (ESKD) (Figure 1). Despite the progressive improvement in patient and kidney survival over the last forty years [7], 10 to 20% of patients still develop ESKD within 10 to 20 years after diagnosis [8,9]. Several studies have reported increased mortality on dialysis in LN-ESKD compared with other causes of ESKD [10,11,12]. Unfortunately, due to the presence of severe comorbidities, not all LN-ESKD patients are admitted to the transplant waiting list. In a large United States Renal Data System (USRDS) cohort, only 9659 (46%) out of the 20,974 patients with LN-ESKD between 1995 and 2014 were admitted to the waiting list for a kidney transplant [13]. The admission to kidney transplantation for ‘healthier LN patients’ alone does not allow for a complete assessment of the potential survival benefit compared to kidney transplants from other causes of ESKD. However, patients who received kidney transplants between 1995 and 2014 in a large cohort in the USRDS had, in comparison to those on the waiting list, a reduced all-cause mortality, particularly for cardiovascular disease and infections [13].

### 2.1. Patient and Graft Survival

Kidney transplant is the modality of renal replacement therapy that guarantees the best quality of life for ESKD in lupus nephritis patients [14]. The USRDS [12,15] and single-center studies [16] that compared the outcome of kidney transplant in patients with LN and matched controls reported that the graft and patient survival rates in LN patients were comparable to those of non-SLE patients. However, between 1998 and 2012, in the Australia and New Zealand cohort, LN-ESKD resulted in worse patient survival, but comparable overall kidney allograft survival and death-censored kidney allograft survival compared to ESKD due to other causes [10]. However, a systematic review of studies with a larger sample size showed that LN was associated with lower patient survival rates in case–control studies, but not in randomized controlled studies (RCTs). Instead, RCTs, but not case–control studies, showed an increased risk of poor graft survival in LN patients [17].

The apparent disparity in results may also be explained by the differences in transplant eras, ethnic composition (especially concerning African Americans), and comparator groups (for example, using diabetic patients as the reference group). Overall kidney transplant patient and graft survival outcomes for lupus and non-lupus recipients were much lower in the US studies [12,15] than in the Australia and New Zealand cohort [10].

The importance of the timing of kidney transplants in patients with LN and kidney failure is debated. Recently, data from USRDS found that graft failure was 16% for LN patients who waited <3 months on dialysis versus 27–30% for those who stayed on dialysis >12 months. However, patients who waited longer for kidney transplants had less favorable sociodemographic, ethnic, and clinical factors, suggesting that these patients probably had active disease or important comorbidities at the time of transplant [18]. Thus, candidates with active lupus and/or those with significant iatrogenic comorbidities should be advised to wait no more than 12 months before transplantation, while patients with a long course of inactive disease can be evaluated for pre-emptive transplant, particularly if a living donor is available.

### 2.2. Recurrence of SLE and LN After Kidney Transplantation and Outcomes and Therapy


(a)Recurrence of SLE


It has been reported that a quarter of SLE patients experienced a disease flare during dialysis, with frequent hematologic manifestations [19]. However, few data are available about the rate and the severity of SLE flares after kidney transplant. Among forty-three lupus patients followed for 8.3 years after kidney transplantation, four (9.3%) developed extrarenal lupus activity requiring an increase in immunosuppression. The manifestations were hematologic in two patients, acute pericarditis in one, and generalized arthralgia with bullous lupus in the last patient. All patients had low serum complement levels and two had anti-dsDNA antibodies [20]. In another study, among 93 patients with LN who underwent kidney transplantation, 11 (11.8%) experienced SLE flares during a follow-up of 76.9 months. Of them, four showed LN recurrence on the graft and four showed hematologic flares. Patients who developed flares had significantly higher anti-dsDNA antibodies both before and after transplantation [21]. In our own experience, mild extrarenal SLE flares characterized by malar rash, arthralgias, and fever occurred in seven out of thirty-five (20%) patients and were treated successfully with a short course of steroids [16].

(b)Recurrence of lupus nephritis and outcomes and therapy

The time of LN recurrence after a kidney transplant is unpredictable. Recurrence can develop during the first months or even 16 years after a kidney transplant [22]. The mean time to recurrence was 4.5 ± 3.7 years in the study by Burgos [23] and was 5.1 ± 4.9 years post-transplantation in a Chinese report [24]. The time of recurrence does not seem to have changed in transplants conducted in different decades. Among 99 Brazilian LN patients, of whom 46% received a graft biopsy in two different periods (before 2009 and from 2009 onwards), the mean time of recurrence was similar between the two groups [25].

The reported data about the incidence of graft recurrence in LN are extremely variable, ranging from 1% to 40% [26,27] depending on the indication for kidney biopsies (protocol biopsies vs. serial biopsies) and their histological analysis. More recently, the extensive use of immunofluorescence and electron microscopy in evaluating graft biopsy has allowed the detection of LN recurrence in around 11–30% of cases [23,26,28]. In our own experience, graft biopsy was performed for clinical reasons, such as deterioration in graft function, proteinuria, and/or active urine sediment. LN recurrence was histologically documented in 8.6% of patients: one patient had class V, one class III, and the last had class IV LN. Recurrences were associated with some extrarenal manifestations and/or the alteration of the immunological tests. Therefore, we cannot rule out the possibility of recurrence in some milder histological forms [16]. In the series of studies by Burgos, among 177 LN transplanted patients, recurrence was documented in 11.3%; the rate of recurrence was 13.9% in African American patients in comparison to 6.9% in Caucasian patients, although the difference was not significant. The histological lesions on the grafts with recurrence were of a mesangial type in 60% of cases, compared with proliferative or membranous lesions in the native kidneys [23]. In a small cohort of Chinese renal transplant recipients with LN who were followed for 20 years, the rate of recurrence was 18.8%, mainly due to class I or II LN [24]. Recently, Contreras et al., by reviewing the United Network for Organ Sharing (UNOS), found a prevalence of recurrence of LN of 2.44% in 6850 patients who received a kidney allograft between 1987 and 2006. However, in the UNOS database, histological classes of recurrent LN were not available; therefore, it is likely that only the more severe forms were included [22]. Out of the 167 recurrent LN patients, 82 (3.19%) were diagnosed before January 1996 and 85 (1.98%) after December 1995 (*p* = 0.0016). Patients who developed recurrence were more frequently black and non-Hispanic, more frequently received deceased-donor kidneys with less compatibility (high levels of HLA-A locus and HLA-B locus mismatch), and had a high frequency of increased panel-reactive antibodies [22]. Similarly, the biopsy-proven LN recurrence rate after transplantation was 2.3% in the contemporary Australia and New Zealand cohort after a median of 4.6 years [10]. With a different approach, Norby et al., in a cross-sectional study, reported a relapse of LN in 54% of 41 Caucasian transplanted patients. Of them, thirty-eight received a protocol biopsy and three a biopsy for clinical reasons. Graft biopsies were performed after a mean duration of transplant of 8 + 4 years. Ten out of the twenty-two recurrent patients had class I, seven had class II LN, and three patients (13.6%) had proliferative forms of LN. The SLEDAI (Systemic Lupus Erythematosus Disease Activity Index) was low and did not differentiate recurrent from nonrecurrent patients, confirming the prevalence of less severe histological forms. Signs of interstitial fibrosis and tubular atrophy were present in 83% of graft biopsies, both in recurrent and in nonrecurrent biopsies [29].

Several factors may predict the development of LN recurrence, including being of black ethnicity, female gender, and younger age, and having anti-phospholipid antibodies and poor compliance with immunosuppression [28,30]. African American ethnicity was found to be associated with a shorter time-to-event for recurrent LN in kidney transplant patients [23]. In a study, lupus anticoagulant was found more frequently in the patients with recurrence; recurrence of LN was also associated with living donation [29]. An increased anti-dsDNA antibody level before transplantation [21] and hypocomplementemia after renal transplantation seemed to be associated with an increased risk of LN recurrence [31]. In a multivariate analysis, being of non-Hispanic black race, female gender, and having an age < 33 years each independently increased the odds of recurrent LN in the largest reported cohort of recurrent LN in kidney transplant patients [22].

Based on the available data, it does not seem that graft recurrence of LN has a strong impact on allograft or patient survival [22,23]. An incidence of 2 to 11% over 5 and 10 years after transplantation is the reported graft loss due to recurrence [22,26,32]. Contreras reported that only 7% of graft failures in LN patients with recurrence could be attributed to recurrence compared to 43% due to rejection.

The management of LN recurrence after kidney transplantation remains challenging, with various immunosuppressive strategies being employed based on disease severity and graft function. Similar to what is in use in non-transplant patients, renin–angiotensin–aldosterone system blockade is used to reduce proteinuria and preserve graft function. The intensification of immunosuppression should be reserved for cases characterized by proliferative forms, the impairment of kidney function, and for extrarenal life-threatening SLE, to avoid an increase in infectious risk [33]. Pattanaik et al. [34] suggested the treatment of LN relapses with prednisone bursts alongside modifications of maintenance immunosuppression. This approach was used in five out of six recurrent LN patients (three methylprednisolone pulses of 500 mg each) with an increase in mycophenolate mofetil dosage. The therapy resulted in the improvement of renal function and a reduction in proteinuria in most cases [24]. Vale et al. [35] described the successful treatment of a graft LN recurrence using high-dose methylprednisolone (0.5 g for three days) followed by cyclophosphamide as an induction regimen. Cyclophosphamide, with the discontinuation of the antimetabolite in treatment, is suggested in cases that present with the rapid deterioration of kidney function associated with diffuse crescents at graft biopsy, and in all cases with severe extrarenal SLE disease [36]. Martizez Lopez et al., in one transplanted patient undergoing treatment with tacrolimus and mycophenloate mofetil, treated the graft LN recurrence with rituximab leading to recovery of the graft function [37]. Rituximab was also used in cases of extrarenal SLE reactivation after kidney transplant [21]. While supporting evidence remains limited, anti-CD 20 therapy may offer an alternative for LN patients with poor responses to conventional approaches, warranting further investigation into its optimal dosing and long-term outcomes.

### 2.3. Conclusions

Kidney transplantation is the most appropriate treatment for lupus patients with ESKD. The patient and graft survival seem to be similar in patients with LN than in standard kidney transplant recipients, but the results may be influenced by the timing of kidney transplantation and choice of immunosuppression in patients with severe comorbidity caused by previous therapies aimed at controlling lupus disease. One main concern is the fear of recurrence after transplantation. Graft survival is significantly worse in recurrent patients than in those without recurrence or rejection. However, only a minority of kidney transplant recipients have lost their allograft because of recurrence. We feel that in patients with the recurrence of LN after kidney transplant, the intensification of immunosuppression should be reserved for the cases characterized by proliferative forms and the impairment of kidney function to avoid an increased risk of infection.

## 3. Kidney Transplant in ANCA-Associated Vasculitis (AAV)

ANCA-associated vasculitides (AAVs) are a rare but heterogeneous group of systemic small-vessel vasculitis, characterized by the necrotizing inflammation of small blood vessels. AAV includes granulomatosis with polyangiitis (GPA), microscopic polyangiitis (MPA), eosinophilic granulomatosis with polyangiitis (EGPA), and forms limited to the kidney called renal-limited vasculitis (RLV) [38,39]. The clinical manifestations of AAV can vary from isolated and indolent single-organ involvement to life-threatening disease. Among the organs involved, the kidney is one of the most severely affected. Except for EGPA, in which kidney involvement occurs in 20–30% of cases [40], in the other forms it develops in 50 to 90% of patients. Kidney disease is characterized by rapidly progressive kidney function deterioration, microscopic hematuria, various degrees of proteinuria, and extra-capillary necrotizing lesions at kidney biopsy (Figure 2) [41,42]. In recent decades, a better understating of the pathogenesis of AAV and the development of more effective therapeutic approaches have resulted in a significant improvement in patient survival [43]. Nonetheless, a large meta-analysis of observational studies including 3338 AAV patients reported a 2.7-fold increase in mortality among these patients compared with the general population [44]. Recent reports also estimate that a quarter of individuals with AAV and glomerular involvement, ANCA-associated glomerulonephritis (ANCA-GN), eventually develop ESKD within five years of diagnosis [45,46,47,48].

### 3.1. Patient and Graft Survival

Only 14% to 22% of patients with ESKD due to ANCA-GN eventually receive a kidney transplant [49,50,51]. The low number of kidney transplants can be partly explained by the older age at diagnosis of ANCA-GN and the frequent comorbidities secondary to the disease and its therapy. The outcomes of kidney transplants in patients with ANCA-GN remain controversial. USRDS reported that among patients with primary or secondary glomerulonephritis in ESKD, those with ANCA-GN had the lowest allograft failure rates together with IgA nephropathy [52]. Single-center studies have also reported that the long-term patient and graft survival of ANCA-GN was similar to that of a matched control group [53,54], despite there being higher infectious complications in the ANCA-GN group. In 558 kidney transplantations in ANCA-GN patients included in 12 renal European registries, patient and graft survival at 10 years were 74.8% and 63.7%, respectively [51]. Other reports have shown that kidney transplant in ANCA-GN was associated with a 70% reduction in the risk of death, particularly from cardiovascular diseases [55]. However, a comprehensive search of PubMed, Scopus, and Embase databases concluded that patients with ESKD and vasculitis undergoing kidney transplantation are at elevated risk of mortality and postoperative infection compared to patients without AAV [56]. Two single-center retrospective cohort studies reported that kidney transplantation in patients with ANCA-GN is associated with a high incidence of post-transplant cancer [57,58].

The timing of kidney transplants in ANCA-GN ESKD patients is critical. Several studies have warned against the high risk of recurrence and mortality in patients who received the transplant while ANCA vasculitis was still active [59,60,61].

The Kidney Disease Improving Global Outcomes (KDIGO) guidelines [62] and the Canadian Society of Transplantation [63] have recommended that patients should be in remission for about 12 months before proceeding with kidney transplantation. ANCA positivity alone is not a contraindication for transplantation.

### 3.2. Recurrence of AAV and ANCA-GN After Kidney Transplantation and Outcomes and Therapy

The post-transplant recurrence rate for ANCA-GN ranged from 9 to 36.8% [53,64,65,66] with an estimated 17% of transplanted patients experiencing recurrence in a pooled analysis [64]. This wide range was largely due to the varying criteria used for diagnosing recurrence, the different periods in which allografts were performed, and the diversity of the immunosuppressive regimens employed. In a review of 428 kidney transplant recipients, Marco et al. [54] found that ANCA-GN recurrence occurred in 47 patients, corresponding to a recurrence rate of 11%, which equaled 0.006–0.08 relapse patient-years. Recurrence may have developed at any time after transplantation, ranging from 0.5 to 109 months [54], with an average time of 31 months [64].

Isolated extrarenal involvement—including the upper respiratory tract, the lungs, the gut, the skin, the joints, and the eyes—occurred in 40% of cases [64,67]. However, around 60% of recurrences involved the transplanted kidney, either alone or in combination with other organs. Clinical signs that may suggest graft recurrence include microscopic hematuria and proteinuria. These signs are often associated with or followed shortly by the deterioration of graft function [68]. The histologic picture of the graft biopsy is similar to that observed in the native kidney. It is characterized by pauci-immune focal or diffuse extra-capillary necrotizing glomerulonephritis. In the Dutch Transplantation Vasculitis (DUTRAVAS) study, ten graft biopsies were classified according to Berden et al. [68]. Five grafts exhibited a focal pattern, four had a mixed pattern, and one displayed a crescentic pattern [68,69].

The relationship between ANCA positivity and AAG recurrence post-transplantation remains debated. It has been reported that persistently elevated ANCA levels, or a rapid increase in ANCA titers, particularly PR3-ANCA, are risk factors for post-transplant recurrence in AAG patients [60,67]. Studies involving large populations have shown that patients with ANCA positivity at the time of transplant are more likely to experience post-transplant AAG recurrence compared to those with normal ANCA levels (17% vs. 5%, respectively) [54]. Additionally, some studies suggest that higher ANCA titers at the time of kidney transplant may lead to earlier recurrence [70,71]. However, this result was not confirmed by other studies, which found that the ANCA pattern, the duration of the original disease, the duration of dialysis, treatment with cyclosporine, the source of donors, and the clinical parameters did not influence the risk of recurrence [53,61,72].

Based on the few available studies and their contradictory results regarding ANCA value in predicting transplant recurrence [73], it seems reasonable not to exclude or delay kidney transplant in patients with complete clinical remission and ANCA positivity [60]. In any case, awaiting more solid evidence, a cautious approach based on the candidate’s characteristics is recommended. Routine ANCA level monitoring before and after transplantation is mandatory [60].

The impact of ANCA-GN recurrence after kidney transplantation is debated. Reviewing data from the Australia and New Zealand Dialysis and Transplant Registry, Briganti et al. reported a 10-year graft loss rate of 7.7% in kidney transplant recipients with relapsing ANCA-GN [74]. However, other studies reported worse outcomes, with early transplant loss in more than 30% of the recurrent patients [69]. Independent predictors of AAV recurrence-related transplant failure include graft involvement and being of male gender [69].

The recurrence of AAV after kidney transplantation requires prompt and effective treatment to prevent graft loss. The therapy in use for post-transplant relapsing ANCA-GN includes methylprednisolone pulses, intravenous cyclophosphamide, and plasmapheresis with a concomitant reduction in anti-rejection drugs [33]. In our experience, seven patients with recurrence were treated with three methylprednisolone pulses (500 mg/day each) followed by oral prednisone at 0.5 mg/kg/day, which was associated with oral cyclophosphamide at 2 mg/kg/day for 2–12 months in five of them. During cyclophosphamide therapy, all the other anti-rejection immunosuppression therapies were withdrawn in three patients; in the other two patients, mycophenolate mofetil and azathioprine were withdrawn, while tacrolimus and cyclosporine were continued unchanged. In three patients, serum creatinine returned to pre-recurrence levels. More recently, rituximab has emerged as a successful therapeutic option for remission induction in recurrent ANCA-GN post-transplant [58,60,75,76,77,78]. Murakami et al. demonstrated that rituximab effectively induced remission in patients with recurrent ANCA-GN, highlighting its role in B-cell depletion and disease control [77]. Similarly, Silva et al. reported a case of ANCA-GN recurrence post-transplant which was successfully managed with rituximab, resulting in disease remission and the preservation of graft function [58]. Another case from Silva et al. described a patient treated with methylprednisolone (500 mg for three consecutive days) followed by rituximab (1 g in two doses, two months apart), which led to a decrease in ANCA levels, improvement in proteinuria, and the normalization of urinary sediment, with stable graft function maintained at one-year follow-up [58]. These findings support rituximab as a viable option for treating recurrent ANCA-GN post-transplant, although further studies are needed to optimize dosing strategies and long-term outcomes [58,69]. No studies compared the efficacy of cyclophosphamide to rituximab for the treatment of recurrent ANCA-GN; however, it is advisable to avoid cyclophosphamide in patients who have already received high doses of this drug in the past. The successful use of eculizumab in 5-year-old patients with immediate AAG recurrence after KT was reported [79].

### 3.3. Conclusions

Despite complex comorbidities and higher risks of infectious and recurrent disease, evidence suggests that kidney transplantation remains the best option for ESKD patients with ANCA-GN [60]. Given the heterogeneity of the disease and the limited amount of available data, individualized approaches are necessary when considering inclusion in the transplant waiting list. International guidelines recommend delaying transplantation until one year of complete clinical remission and advise caution in selecting patients who have undergone intense immunosuppressive therapy [59,62,80,81]. ANCA-GN recurrence occurs in up to 11% of kidney transplant recipients and is frequently associated with severe complications and premature transplant loss [54]. Worsening graft function, abnormal proteinuria, and microscopic hematuria are indicators of graft recurrence. Graft biopsy should be performed as early as possible as the prompt initiation of treatment may improve response rates and outcomes. To date, there is no standardized treatment for recurrent ANCA-GN [27,53,68].

## 4. Henoch–Schönlein Purpura (HSP)

IgA vasculitis, formerly Henoch–Schönlein purpura (HSP), is a systemic vasculitis of the small vessels with IgA1-dominant immune deposits. It is the most frequent systemic vasculitis in children, being rarer and having a more severe presentation and outcome in adults. Clinical manifestations include cutaneous purpura, arthritis, acute enteritis, and glomerulonephritis (Figure 3). Gastrointestinal and kidney involvement represent the primary cause of morbidity and mortality in adults. Kidney involvement occurs in 32% to 85% of adults with HSP [82,83,84].

In two large cohorts of adults with HSP, 30% of patients had chronic kidney disease (CKD) at baseline [85,86]. In 250 HSP patients followed for 14.8 years, patient survival was only 74%, with carcinoma as the first cause of death. At the last observation, only 20% of patients were in remission, 27% had CKD (creatinine clearance <30 mL/min in 13%, and <50 mL/min in 14%), and 11% of patients reached ESKD [86].

A systemic review of available studies has reported that the use of corticosteroids, cyclophosphamide, mycophenolate, heparin, tacrolimus, or antiplatelet agents to prevent persistent kidney disease in children with HSP is questionable, because there is little evidence of their efficacy and they have high risks of toxicity and side effects [87,88]. The only randomized study in HSP adult patients with kidney involvement showed that cyclophosphamide did not improve patient and kidney survival. Some recent experiences suggest the efficacy of rituximab, but results should be confirmed in larger studies [88].

### 4.1. Patient and Graft Survival

The largest cohort recently reported by the Scientific Registry of Transplant Recipients (SRTR) included 371 pediatric and adult patients with HSP who received a kidney transplant between 2005 and 2021. Their outcome was compared with that of 1113 non-HSP-matched recipients. Patient survival at 5 years was higher for HSP patients at 96.0% compared to 93.9% of controls (*p* = 0.020). However, death-censored graft survival was similar between the groups at 5 years, at 88.0% versus 85.3% (*p* = 0.223). No significant differences were observed in the survival rates between pediatric and adult patients with HSP, or when comparing subgroups of pediatric and adult HSP patients to their matched controls [89]. Other studies involving small numbers of patients show varying outcomes with a Japanese cohort of 21 patients with HSP and 42 controls showing a 15-year survival rate of 100% for HSP vs. 97.6% for controls (*p* = 0.22). The 5-, 10-, and 15-year graft survival rates were 95.2, 90.5, and 81% in IgA vasculitis and 100, 90.5, and 88.1% in the controls, respectively [90]. Previous studies have reported that kidney and patient survival rates in HSP transplant patients are generally good, with no significant differences from control groups [91,92,93]. The optimal timing of transplantation for HSP patients in renal replacement therapy is still uncertain. Concerns have been raised about pre-emptive transplant potentially leading to worse outcomes in children [94]. Meulders et al. reported that patients with a rapidly progressing disease in the native kidneys may have an increased risk of recurrence post-transplant. In such cases, waiting for a period of disease quiescence (when the disease is stable) may be recommended before transplanting [95].

### 4.2. Recurrence of HPS After Kidney Transplantation and Outcomes and Therapy

The rarity of the disease is a major limitation in assessing the recurrence rate of HSP in kidney transplantation. In the largest and most recent cohort reported in the literature, HSP recurrence was significantly higher in adults than in children (29.7% vs. 13.0%, *p* = 0.011) [89].

This result contrasts with previous studies that reported a higher recurrence rate in children than in adults, with there being more than 60% of recurrences at 15 years after transplantation [94,96].

The rate of HSP recurrence in adults is highly variable. The percentage of clinical recurrences ranges from 15 to 42% in HSP patients undergoing biopsies for clinical reasons or after a long kidney transplant observation [90,91,93]. Three out of the twenty HSP transplanted patients described by Han et al. had recurrence, with the 5- and 10-year cumulative rates of recurrence being 7.7% and 15.4% [93]. In surveillance kidney biopsies, the recurrence rate was reported to be higher, with more rapid occurrence. Thervet et al. performed 66 graft routine biopsies (protocol surveillance and clinical biopsies) in 18 kidney biopsies of 13 HSP patients [97]. Recurrence occurred in 69% of patients and 61% of grafts after a mean observation period of 24 months. Clinical manifestations of recurrence were absent in all but one case, but mesangial IgA deposition at immunofluorescence confirmed the histological diagnosis of recurrence.

In our single-center experience, which included 19 grafts in 17 HSP patients whose outcome was compared with that of a well-matched control group of 38 patients, the indications for graft biopsy were graft dysfunction, and/or proteinuria, and/or hematuria. HSP recurrence was documented by immunofluorescence in eight grafts (42%) of seven patients 30 + 41 months after transplant, accounting for a relapse rate of 0.05/patient/year. Microscopic or macroscopic hematuria was the first clinical manifestation that heralded HSP recurrence. A significantly higher rate of recurrences of HSP nephritis was observed in our patients with crescentic glomerulonephritis in the native kidney [91]. These data were confirmed by the study by Soler et al. [98]. No other differences were observed between recurrent and nonrecurrent patients. In a pooled analysis performed by Han et al., the risk of recurrence was higher in living-related donor transplants than in the recipients of grafts from deceased donors [93]; this difference was not confirmed by other studies [89,91]. Kanaan et al. collected data from six transplant centers in Belgium and France and reported the lowest recurrence rates in HSP [99]. In 46 HSP kidney transplant recipients with a follow-up of >3 years, the actuarial risk for clinical recurrence in the first graft was 2.5% at 5 years and 11.5% at 10 years. Only two out of the five patients with recurrences showed systemic signs of HSP. The severity of the disease at presentation and the type of immunosuppression after transplantation did not affect recurrence [99].

The percentage of graft loss due to recurrence has been reported to range between 0 and 21% [91,95,100,101]. No difference in the rate of loss due to recurrence was found between different donor sources [92,97]. The impact of recurrence on graft survival largely depends on whether the diagnosis was based on clinical or histological grounds. In the surveillance graft biopsies reported by Thervet E et al. [97], none of the grafts lost were due to recurrence of the disease. Hasegawa found that only 5 patients out of 15 with a histological diagnosis of HSP recurrence developed proteinuria and hematuria during the follow-up, and two of them lost the graft due to recurrence [96]. Three of the five recurrent graft recipients described by Kanaan et al. lost their first graft due to HSP recurrence with a risk of 2.5% at 5 years and of 7.5% at 10 years [99]. Samuel et al. [92], using the UNOS database, reported graft failure due to recurrent disease in 13.6% of HPS patients versus 6.6% in the control group, with the difference being statistically significant. Patients with recurrence had a lower graft and patient survival at 1, 3 and 5 years than those without recurrence [89].

Various therapeutic approaches have been used in the treatment of HSP recurrences, including methylprednisolone pulses and immunosuppressive agents, though the results have been unsatisfactory. Methylprednisolone pulses were most frequently employed in severe clinical and histological forms with contrasting results. In our patients, this treatment did not improve graft survival, although it was successful in three cases of clinical recurrence, as reported by Mayuko Kawabe et al. [90]. A case of a positive outcome following methylprednisolone pulses and tonsillectomy for HSP recurrence was also reported in a Japanese patient [102]. Lee et al. [103] demonstrated the potential usefulness of therapeutic plasma exchange (TPE) for IgA vasculitis recurrence after transplantation, though no additional data were available. The utility of rituximab for recurrent post-transplant HSP is not well established, though some data have suggested it may benefit children with HSP [104,105,106].

### 4.3. Conclusions

Despite the rarity of the disease, all the available data suggest that patient and graft survival after kidney transplant are good and are not different from that of matched groups. Although no guidelines are available, disease quiescence before transplant may be suggested in patients with rapidly progressive disease in the native kidney. The rate of recurrence is high in surveillance biopsies, but their outcome is generally good. Instead, clinical recurrences are associated with reduced graft survival, which depends on some factors including the duration of the observation and the severity of the clinical and histological presentation of recurrence. Microscopic hematuria heralds graft recurrence and is an indication for graft biopsy. Methylprednisolone pulses may be suggested for crescentic forms, although their efficacy has been demonstrated only in some cases.

## 5. Summary and Conclusions—Table 1

In essence, there is no reason to exclude patients affected by the aforementioned systemic diseases from transplantation. Quality of life and survival rates are better after kidney transplantation than in patients who remain on dialysis.

**Table 1 jcm-14-02592-t001:** Outcomes of kidney transplantation in patients with lupus nephritis (LN), ANCA-associated glomerulonephritis (ANCA-GN), and Henoch–Schönlein purpura (HSP). This table compares kidney transplant, admission rates, transplantation timing, patient survival, graft survival, recurrence rates (in clinical and surveillance kidney biopsies, KBs), predictors, and the impact of recurrence in the three diseases. MMP, mycophenolate mofetil; Cyc, cyclophosphamide; RTX, rituximab.

	LN	ANCA-GN	HSP
Admission to kidney transplantation	46%	12–24%	Unknown
Timing of transplantation	No more than 12 months in patients with active LN	Wait 12 months for remission	Wait for remission in patients with rapidly progressive disease
Patient survival	Reduced in deceased donor recipients	High early mortality	Better than controls
Graft survival	Reduced in deceased donor recipients	Good after one year; increased cancer risk	Similar to controls
Recurrence rate in clinical kidney biopsy	2.5–11%	9–36.8%	15–42%
Recurrence rate in surveillance kidney biopsy	30–56%	Not performed	69%
Predictors of recurrence	Black non-Hispanic ethnicity, females, age < 33 years	ANCA positivity: debated; no other predictors	Crescentic glomerulonephritis in the native kidney
Impact of recurrence	2–11%: No negative impact	Up to 30%: Impact undefined	13.6% vs. 6.6%: Negative impact
Treatment of recurrence	Only for class III or class IV MMP pulses, Cyc, RTX	MMP, Cyc, RTX, plasma exchange	No effective therapy

The two main issues that may arise in patients with systemic diseases are the risk of immunosuppression in individuals severely compromised by the underlying disease and the risk of recurrence. In patients who have received significant immunosuppression to combat the original disease, a period of several months on dialysis may be advisable to allow for the clearance of the harmful effects of corticosteroids or other immunosuppressive agents.

The risk of recurrence is not higher in systemic diseases than that which is expected for primary glomerular diseases. At present, there are no specific therapies aimed at reducing the risk of recurrence. However, in the near future, new complement modulators or specific monoclonal antibodies targeting certain pathogenic elements may further improve kidney transplant outcomes in systemic diseases and reduce the risk of recurrence. Several complement inhibitors have proved to be able to reduce proteinuria, and will probably be effective at slowing the progression of the disease [107]. New monoclonal antibodies may target both the B-cells and T-cells involved in the pathogenesis of systemic diseases and also the factors needed for their activation and survival.

## Figures and Tables

**Figure 1 jcm-14-02592-f001:**
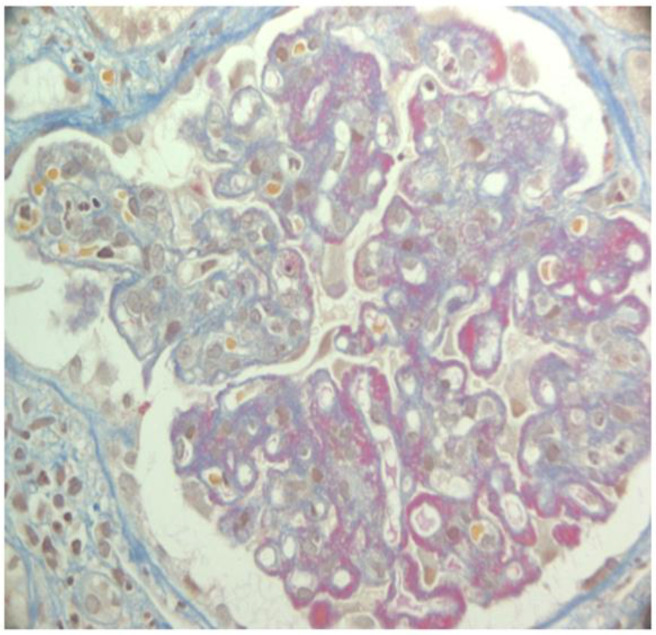
Lupus nephritis class IV. AFOG’s trichrome. Glomerulus with intracapillary proliferation and diffuse thickening of capillary walls due to the presence of massive wire loops.

**Figure 2 jcm-14-02592-f002:**
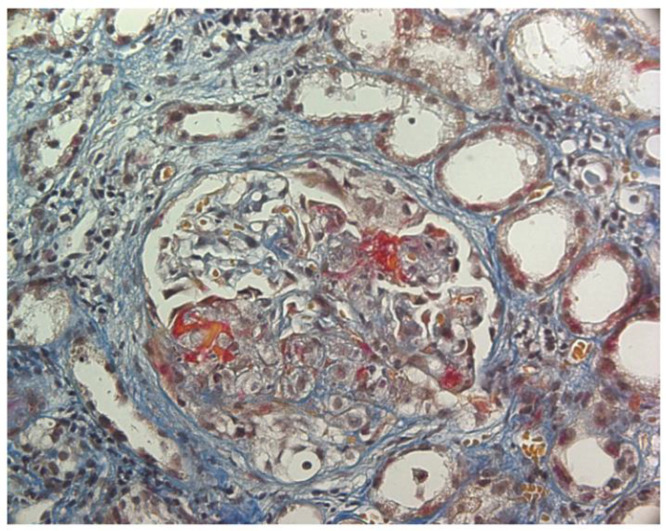
ANCA-associated glomerulonephritis. AFOG’s trichrome stain. Glomerulus with extra-capillary proliferation and necrotizing lesions of the tuft.

**Figure 3 jcm-14-02592-f003:**
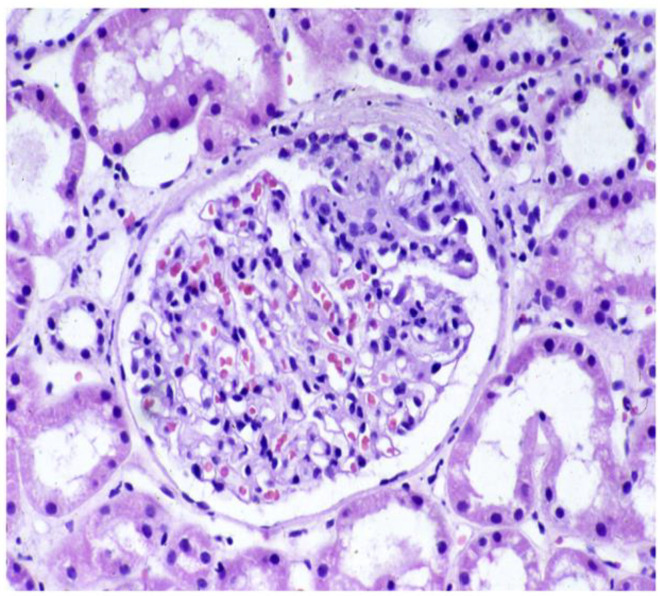
Shönlein–Henoch purpura. PAS stein. Glomerulus with segmental increase in mesangial cells and matrix and adhesion to the Bowman capsule.

## Data Availability

Not applicable.

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
