# Peer review of "The Recurrence of Systemic Diseases in Kidney Transplantation"

_jcm, 2025, doi:10.3390/jcm14082592_

Round 1

Reviewer 1 Report

Comments and Suggestions for Authors

The manuscript provides a well-structured narrative review of the recurrence of lupus nephritis (LN), ANCA-associated vasculitis (AAV), and Henoch-Schönlein purpura (HSP) following renal transplantation. The topic is highly relevant and underinvestigated. The review covers the timing, frequency, severity, and treatment of recurrence for these three diseases in a thorough way. I have no major issues, but a few areas could be improved, as listed below.

  1. The manuscript uses varying terms (e.g., "ANCA-associated glomerulonephritis" vs. "AAG" vs. "AAV;" "HSP" "IgA vasculitis”) without clear standardization. Please maintain consistent terminology throughout.
  2. The treatment options for recurrence are mentioned but lack specificity regarding efficacy, dosing, or supporting evidence. For instance, the statement "good results with rituximab have been reported" (Section 3.2) is noted but specific studies or outcomes should be cited to substantiate this claim.
  3. Please consider adding a table comparing the impact of recurrence on graft survival across LN, AAV, and HSP and/or a table summarizing recommended therapies, doses, and evidence levels for each disease. This data visualization could significantly enhance the manuscript’s utility for clinicians.
  4. Minor errors exist, such as "HPS" instead of "HSP" (L 321).

Author Response

The manuscript provides a well-structured narrative review of the recurrence of lupus nephritis (LN), ANCA-associated vasculitis (AAV), and Henoch-Schönlein purpura (HSP) following renal transplantation. The topic is highly relevant and underinvestigated. The review covers the timing, frequency, severity, and treatment of recurrence for these three diseases in a thorough way. I have no major issues, but a few areas could be improved, as listed below.

  1. The manuscript uses varying terms (e.g., "ANCA-associated glomerulonephritis" vs. "AAG" vs. "AAV;" "HSP" "IgA vasculitis”) without clear standardization. Please maintain consistent terminology throughout.

For ANCA associated vasculitis, we use both the acronym “AAV” , while for ANCA-associated glomerulonephritis, we use “ANCA-GN”, to differentiate between the systemic disease and renal involvement (as specified in L253). IgA vasculitis is formerly called Henoch-Schönlein purpura (as specified in L372), so both terms can be used. Throughout the text, we consistently use the acronym “HSP”.

2. The treatment options for recurrence are mentioned but lack specificity regarding efficacy, dosing, or supporting evidence. For instance, the statement "good results with rituximab have been reported" (Section 3.2) is noted but specific studies or outcomes should be cited to substantiate this claim.

We have added the few available data from the literature regarding specific treatment options for recurrence in LN and ANCA-GN (L201 and L329).  For HSP, we confirm the previous reported data. Due to the rarity of these diseases, no randomized trials have been conducted to evaluate the efficacy of therapy in recurrence, and no evidence-level data are available.

3. Please consider adding a table comparing the impact of recurrence on graft survival across LN, AAV, and HSP and/or a table summarizing recommended therapies, doses, and evidence levels for each disease. This data visualization could significantly enhance the manuscript’s utility for clinicians.

Thank you for this suggestion, we have added a Table (Table 1).

  1. Minor errors exist, such as "HPS" instead of "HSP" (L 321).

Thank you, we have corrected the identified errors

We thank Reviewer 1 for their feedback and the careful consideration of our work.

Reviewer 2 Report

Comments and Suggestions for Authors

Authors submitted an interesting review about the possible impact of selected diseases: lupus nephritis, ANCA vasculitis and HSP reccurrence on patient's and graft function. Since these immune mediated diseases may require targeted treatment, and after kidney transplantation many other disorders should be taken under consideration in differential diagnosis, the appropriate managment of kidney transplant patients with concomittant diseases is extremely difficult.

Comments:

1) please consider using 'kidney' instead of 'renal';

2) L12-14: I suggest to change that sentence, KTx is not ideal for every patients/any disease leading to kidney failure (i.e. kidney cancer),

2) L22: I suggest to add 'primary' focal segmental glomerulosclerosis,

3) please explain all abbreviations, i.e. 'ANCA' (L20), or decide if you use full name or abbreviation (i.e. L223, L315),

4) please use appropriately 'anti-dsDNA antibodies' (L111-112),

5) L223: I suggest to use 'primary' vs secondary FSGS,

6) L303: AAG/AAV recurrence,

7) L331: please change % position into: 93.9 % of controls,

8) L343: Meulders et al.

9) L389: recurrence of the disease/HSP,

10) L405-406: I suggest to use TPE,

11) L418: methylprednisolone.

Author Response

Authors submitted an interesting review about the possible impact of selected diseases: lupus nephritis, ANCA vasculitis and HSP reccurrence on patient's and graft function. Since these immune mediated diseases may require targeted treatment, and after kidney transplantation many other disorders should be taken under consideration in differential diagnosis, the appropriate managment of kidney transplant patients with concomittant diseases is extremely difficult. Comments:

1) please consider using 'kidney' instead of 'renal';

2) L12-14: I suggest to change that sentence, KTx is not ideal for every patients/any disease leading to kidney failure (i.e. kidney cancer),

2) L22: I suggest to add 'primary' focal segmental glomerulosclerosis,

3) please explain all abbreviations, i.e. 'ANCA' (L20), or decide if you use full name or abbreviation (i.e. L223, L315),

4) please use appropriately 'anti-dsDNA antibodies' (L111-112),

5) L223: I suggest to use 'primary' vs secondary FSGS,

6) L303: AAG/AAV recurrence,

7) L331: please change % position into: 93.9 % of controls,

8) L343: Meulders et al.

9) L389: recurrence of the disease/HSP,

10) L405-406: I suggest to use TPE,

11) L418: methylprednisolone.

We thank Reviewer 2 for the positive feedback and your suggestions. We have made all the suggested changes.

Reviewer 3 Report

Comments and Suggestions for Authors

The article's authors have conducted a narrative review entitled “The Recurrence of Systemic Diseases in Renal Transplantation”. The subject of the work is undoubtedly interesting and relevant from the point of view of clinical practice. The authors attempt to discuss three critical diseases – systemic lupus erythematosus (lupus nephritis), ANCA-associated vasculitis and Henoch-Schönlein purpura (IgA vasculitis) – in the context of their recurrence after kidney transplantation.

  1. The article is a valuable contribution to the nephrology literature. Still, it is worth considering several editorial and substantive issues that may improve its coherence, clarity, and clinical usefulness:
  2. Review methodology—Although this is a narrative review, it would be advisable to include a brief description of the methods of literature selection: what time period the searched publications covered, what databases were used, and what types of papers were included. Although the narrative nature of the work does not require the full rigor of a systematic review, adding such a section would significantly increase the clarity and objectivity of the study.
  3. Purpose of the work – I suggest outlining the purpose of the review more clearly in the introduction so that the reader knows from the outset what to expect and why it is worth reading the rest of the content.
  4. Visual elements – considering introducing figures (e.g. diagrams, schemes) could increase the clarity and accessibility of the data, as well as improve the citation of the article.
  5. Summary tables – I recommend introducing at least one table summarizing the most important information regarding the disease entities being compared: recurrence rate, time to recurrence, risk factors, impact on graft survival, etc. This would significantly facilitate quick comparison of data for readers.
  6. Editorial consistency – please review the manuscript carefully for formatting. In places such as lines 176 and 178, there are inconsistencies in font size, which may give the impression of a lack of editorial care.
  7. Extending the conclusions – it is worth considering whether to add a reference to future research directions, potential development perspectives for this topic or general recommendations for clinical practice in the summary. Such an emphasis would give the work additional value and show the authors' broader perspective on the subject.
  8. Bibliography – generally up-to-date and extensive, which should be appreciated. Nevertheless, it is worth checking for duplicates (e.g., items 30 and 33 seem to be a repetition of the same publication). I would also consider the necessity of citing very old sources (pre-1995), unless they refer to sporadic cases or are classic source publications. Where possible, it is worth looking for newer studies.

In summary, the authors have addressed an important and clinically useful topic. The work has the potential to become a valuable source of knowledge for nephrologists and transplantologists, but it still requires refinement in terms of data presentation, content organization, and technical editing. Considering the above comments will contribute to increased transparency and quality of the publication.

Author Response

The article's authors have conducted a narrative review entitled “The Recurrence of Systemic Diseases in Renal Transplantation”. The subject of the work is undoubtedly interesting and relevant from the point of view of clinical practice. The authors attempt to discuss three critical diseases – systemic lupus erythematosus (lupus nephritis), ANCA-associated vasculitis and Henoch-Schönlein purpura (IgA vasculitis) – in the context of their recurrence after kidney transplantation.

  1. The article is a valuable contribution to the nephrology literature. Still, it is worth considering several editorial and substantive issues that may improve its coherence, clarity, and clinical usefulness:
  2. Review methodology—Although this is a narrative review, it would be advisable to include a brief description of the methods of literature selection: what time period the searched publications covered, what databases were used, and what types of papers were included. Although the narrative nature of the work does not require the full rigor of a systematic review, adding such a section would significantly increase the clarity and objectivity of the study.

We add a new paragraph “Materials and Methods” at the end of the Introduction in which we have reported a brief description of the methods used.  (L60)

3. Purpose of the work – I suggest outlining the purpose of the review more clearly in the introduction so that the reader knows from the outset what to expect and why it is worth reading the rest of the content.

As requested, in the new paragraph, we have added some sentences to better explain the purpose of the study.  (L69)

4. Visual elements – considering introducing figures (e.g. diagrams, schemes) could increase the clarity and accessibility of the data, as well as improve the citation of the article.

We have added a histological picture for any of the diseases: Figure 1 for lupus nephritis, Figure 2 for ANCA associated glomerulonephritis, Figure 3 for IgA vasculitis.

5. Summary tables – I recommend introducing at least one table summarizing the most important information regarding the disease entities being compared: recurrence rate, time to recurrence, risk factors, impact on graft survival, etc. This would significantly facilitate quick comparison of data for readers.

We have added a table reporting the requested data (Table1).

6. Editorial consistency – please review the manuscript carefully for formatting. In places such as lines 176 and 178, there are inconsistencies in font size, which may give the impression of a lack of editorial care.

We have corrected the formatting.

7. Extending the conclusions – it is worth considering whether to add a reference to future research directions, potential development perspectives for this topic or general recommendations for clinical practice in the summary. Such an emphasis would give the work additional value and show the authors' broader perspective on the subject.

We have added a paragraph at the end of the paper summarizing the conclusions (L487)

8. Bibliography – generally up-to-date and extensive, which should be appreciated. Nevertheless, it is worth checking for duplicates (e.g., items 30 and 33 seem to be a repetition of the same publication). I would also consider the necessity of citing very old sources (pre-1995), unless they refer to sporadic cases or are classic source publications. Where possible, it is worth looking for newer studies.  

The great majority of the citations refer to new studies. We have removed some old studies.:

-Nyberg, G.; Blohmé, I.; Persson, H.; Olausson, M.; Svalander, C. Recurrence of SLE in Transplanted Kidneys: A Follow-up Transplant Biopsy Study. Nephrol Dial Transplant 1992, 7, 1116–1123.

                -Stone, J.H.; Millward, C.L.; Olson, J.L.; Amend, W.J.; Criswell, L.A. Frequency of Recurrent Lupus Nephritis among Ninety-Seven Renal Transplant Patients during the Cyclosporine Era. Arthritis Rheum 1998, 41, 678–686, doi:10.1002/1529-0131(199804)41:4<678::AID-ART15>3.0.CO;2-7.

                -Nachman, P.H.; Hogan, S.L.; Jennette, J.C.; Falk, R.J. Treatment Response and Relapse in Antineutrophil Cytoplasmic Autoantibody-Associated Microscopic Polyangiitis and Glomerulonephritis. Journal of the American Society of Nephrology 1996, 7, 33–39, doi:10.1681/ASN.V7133.

-Westman, K.W.; Bygren, P.G.; Olsson, H.; Ranstam, J.; Wieslander, J. Relapse Rate, Renal Survival, and Cancer Morbidity in Patients with Wegener’s Granulomatosis or Microscopic Polyangiitis with Renal Involvement. J Am Soc Nephrol 1998, 9, 842–852, doi:10.1681/ASN.V95842